# Enhancing Selective Antimicrobial and Antibiofilm Activities of Melittin through 6-Aminohexanoic Acid Substitution

**DOI:** 10.3390/biom14060699

**Published:** 2024-06-14

**Authors:** Naveenkumar Radhakrishnan, Sukumar Dinesh Kumar, Song-Yub Shin, Sungtae Yang

**Affiliations:** 1Department of Biomedical Sciences, School of Medicine, Chosun University, Gwangju 61452, Republic of Korea; naveens4596@gmail.com (N.R.); sdkumarphd@gmail.com (S.D.K.); 2Department of Cellular and Molecular Medicine, School of Medicine, Chosun University, Gwangju 61452, Republic of Korea; syshin@chosun.ac.kr; 3Department of Microbiology, School of Medicine, Chosun University, Gwangju 61452, Republic of Korea

**Keywords:** antimicrobial peptide, melittin, 6-aminohexanoic acid, leucine, drug-resistant bacteria

## Abstract

Leucine residues are commonly found in the hydrophobic face of antimicrobial peptides (AMPs) and are crucial for membrane permeabilization, leading to the cell death of invading pathogens. Melittin, which contains four leucine residues, demonstrates broad-spectrum antimicrobial properties but also significant cytotoxicity against mammalian cells. To enhance the cell selectivity of melittin, this study synthesized five analogs by replacing leucine with its structural isomer, 6-aminohexanoic acid. Among these analogs, Mel-LX3 exhibited potent antibacterial activity against both Gram-positive and Gram-negative bacteria. Importantly, Mel-LX3 displayed significantly reduced hemolytic and cytotoxic effects compared to melittin. Mechanistic studies, including membrane depolarization, SYTOX green uptake, FACScan analysis, and inner/outer membrane permeation assays, demonstrated that Mel-LX3 effectively permeabilized bacterial membranes similar to melittin. Notably, Mel-LX3 showed robust antibacterial activity against methicillin-resistant *Staphylococcus aureus* (MRSA) and multidrug-resistant *Pseudomonas aeruginosa* (MDRPA). Furthermore, Mel-LX3 effectively inhibited biofilm formation and eradicated existing biofilms of MDRPA. With its improved selective antimicrobial and antibiofilm activities, Mel-LX3 emerges as a promising candidate for the development of novel antimicrobial agents. We propose that the substitution of leucine with 6-aminohexanoic acid in AMPs represents a significant strategy for combating resistant bacteria.

## 1. Introduction

Antimicrobial peptides (AMPs) are derived from a wide range of living organisms and assist in combating invading pathogens [1,2]. These peptides demonstrate broad-spectrum antimicrobial activity, effectively targeting bacteria, viruses, and fungi [3,4]. Unlike conventional antibiotics, which often target specific cellular components or processes, AMPs exert their antimicrobial effects through various mechanisms, including membrane disruption, inhibition of intracellular targets, and modulation of immune responses [5,6]. With the rise of antibiotic-resistant pathogens and given that bacteria are known to have difficulty developing resistance to AMPs, there is a growing interest in AMPs as alternative strategies for combating infectious diseases [7,8,9]. Research efforts are focused on understanding the structure–function relationships of AMPs, identifying new sources, and exploring their therapeutic potential in various medical and biotechnological applications [10,11,12].

Melittin (GIGAVLKVLTTGLPALISWIKRKRQQ), derived from the venom of the European honeybee (*Apis melifera*), is a compelling AMP that has attracted significant attention in scientific research and medical applications [13,14,15]. Recent studies have assessed melittin for its therapeutic potential in various diseases, including rheumatoid arthritis [16,17], chronic pain [18], anti-nociceptive effects [19,20], anti-mutagenic properties [21], anticancer activity [22,23], and radioprotective effects [24]. However, its clinical translation is hindered by its inherent cytotoxicity towards mammalian cells. Melittin contains the leucine zipper motif, and its structure is marked by an amphipathic bent helix due to the presence of proline in the central region [25]. These structural features contribute to its ability to disrupt lipid membranes, leading to cell lysis and death. This mechanism of action makes melittin effective against a broad spectrum of microorganisms and mammalian cells alike [26,27,28]. For example, the leucine zipper can contribute to the ability of melittin to form oligomers or aggregates, which can enhance its cytolytic activity by facilitating membrane disruption and pore formation [29,30]. Therefore, understanding this structural feature is crucial for elucidating the mechanism of action of melittin and its pivotal role in membrane disruption, which underpins its antimicrobial and cytolytic activities.

6-Aminohexanoic acid is an unnatural amino acid characterized by a six-carbon aliphatic chain, comprising an amino group at one end and a carboxylic acid group at the other. Despite sharing the same molecular weight as leucine, 6-aminohexanoic acid offers flexibility and mobility due to its central aliphatic chain (Appendix A). This structural divergence confers unique properties and biological activities on 6-aminohexanoic acid [31,32]. In this study, to modify the leucine zipper motif and to enhance the structural flexibility of melittin, each or all leucine residues at four positions (L6, L9, L13, and L16) of melittin were substituted with 6-aminohexanoic acids (Table 1). To assess the cell selectivity of the peptides, we examined their antibacterial activities against Gram-positive and Gram-negative bacteria, as well as their cytotoxicity against mammalian cells. The secondary structures of melittin and its analogs were analyzed in various environments using circular dichroism, while the interactions of the peptides with biological and/or artificial membranes were investigated using fluorescence spectroscopy. Furthermore, we evaluated the antimicrobial and antibiofilm activities of the peptides against drug-resistant bacteria. We discovered that Mel-LX3, wherein leucine at position 13 was replaced by 6-Aminohexanoic acid, exhibited improved cell selectivity and antibiofilm activity against drug-resistant bacteria.

## 2. Materials and Methods

### 2.1. Materials

Resins of amide-methyl benzhydrylamine (MBHA) and amino acids pre-treated with 9-fluorenyl-methoxycarbonyl (Fmoc) protective groups were acquired from Novabiochem (La Jolla, CA, USA) for peptide synthesis. Several reagents were sourced from Sigma-Aldrich (St. Louis, MO, USA), including 2,2,2-trifluoroethanol, N-phenyl-1-napthylamine (NPN), sodium dodecyl sulfate (SDS), *o*-nitrophenyl-*β*-galactosidase (ONPG), 3,3′-Dipropylthiadicarbocyanine iodide (diSC_3_-5), and calcein. SYTOX green dye was supplied from Thermo Fisher Scientific, South Korea. All the buffers were made by using Milli-Q ultrapure water (Merck Millipore, USA). The bacterial strains utilized in this study comprised *Staphylococcus aureus* (KCTC 1621), *Staphylococcus epidermitis* (KCTC 1917), *Bacillus subtilis* (KCTC 3068), *Escherichia coli* (KCTC 1682), *Salmonella typhimurium* (KCTC 1926), and *Pseudomonas aeruginosa* (KCTC 1637). These strains were acquired from the Research Institute of Bioscience and Biotechnology (KRIBB)’s Korean Collection for Type Cultures (KCTC) of South Korea. Additionally, multidrug-resistant *Pseudomonas aeruginosa* (CCARM 2095) and methicillin-resistant strains of *Staphylococcus aureus* (CCARM 3090) were obtained from Seoul Women’s University in South Korea.

### 2.2. Peptide Synthesis

The Fmoc-based solid-phase method was employed to synthesize all peptides with MBHA resin [33]. Peptide purity was assessed using an RP-HPLC C_18_ column (Vydac, 250 × 20 mm, 15 μm, 300 Å). The molecular masses of the peptides were determined using a triple-quadrupole mass spectrometer equipped with an electrospray ionization liquid chromatography–mass spectrometry (ESI-LC-MS) system (API2000, AB SCIEX).

### 2.3. Circular Dichroism (CD) Spectroscopy

CD spectroscopy was employed to analyze the secondary structure of the peptides under various environmental conditions, as described previously [34,35]. CD spectra were recorded between 190 and 250 nm at a scan speed of 10 nm/min at 25 °C in 10 mM sodium phosphate buffer (pH 7.4), 30 mM SDS micelles, and 50% trifluoroethanol (TFE) using a spectropolarimeter J-715 (Jasco-Tokyo, Japan) equipped with a 0.1 cm long rectangular quartz cell. Three scans were performed for each peptide, and the results were compiled and averaged. Peptides were concentrated to a final concentration of 100 μg/μL. The mean residue ellipticity ([θ]M) was calculated using the formula [θ]M = (θ_obs_ × 1000)/(c × l × n), where θ_obs_ represents the observed ellipticity corrected for the buffer at a given wavelength (mdeg), c is the peptide concentration (μM), l is the path length (mm), and n is the number of amino acids in the peptide sequence.

### 2.4. Antimicrobial Activity

Peptides were assessed for their antibacterial activity against four Gram-positive and four Gram-negative bacteria, including drug-resistant strains, in accordance with Clinical and Laboratory Standards Institute (CLSI) guidelines [36]. The minimum inhibitory concentration (MIC) was determined using a microtiter broth dilution method. All bacterial strains were incubated overnight at 37 °C. Cultured bacteria were then diluted 10-fold in Muller–Hinton Broth (MHB) medium (Difco, Detroit, MI, USA) and incubated for 3 h to reach mid-log phase. Following the establishment of mid-log phase cultures, bacteria were inoculated into sterile 96-well plates containing serially diluted peptides. The MIC was defined as the lowest concentration of peptides that prevented observable turbidity after incubation of the plates at 37 °C for 18 to 24 h.

### 2.5. Hemolytic Activity

The hemolytic activity of peptides was assessed by measuring the release of hemoglobin from sheep red blood cells (sRBCs), as described previously [37]. To create a 4% *v*/*v* erythrocyte dilution, fresh sRBCs were washed with phosphate-buffered saline (PBS), centrifuged, and then resuspended in PBS. Next, 100 μL of serially diluted peptides was added to a sterile 96-well plate, followed by the addition of 100 µL of sRBCs. The plate was then incubated at 37 °C for one hour. After incubation, the plate was centrifuged at 1000 relative centrifugal force (RCF), and the supernatant was transferred to a new 96-well plate. Hemoglobin release was measured at 405 nm using a microplate ELISA reader. A total of 0.1% Triton X-100 treatment was utilized to induce 100% hemolysis, while PBS was employed as the reference for 0% hemolysis. The percentage of hemolytic activity was determined using the formula % Hemolysis = [(Abs in peptide solution − Abs in PBS)/(Abs of 0.1% Triton X-100 − Abs of PBS)] × 100.

### 2.6. Cytotoxicity against RAW 264.7 Cells

We conducted the MTT assay (3-(4,5-dimethylthiazol-2-yl)-2,5-diphenyltetrazolium bromide) dye reduction experiment against RAW 264.7 macrophage cells to evaluate the cytotoxicity of the peptides, following previously published methods [38,39]. RAW 264.7 cells were cultured for 24 h in the presence of 5% CO_2_ at 37 °C after seeding them in 96-well plates (2 × 10^4^ cells/well). Subsequently, the peptides were treated at increasing concentrations and allowed to interact with the cells for 48 h. Following the incubation period, 20 μL of MTT solution (5 mg/mL in PBS) was added to each well, and the plate was further incubated at 37 °C for 4 h. The absorbance was then measured at 550 nm using a microplate ELISA reader after dissolving the formazan crystals in dimethyl sulfoxide (DMSO). Treatment with 0.1% Triton X-100 served as a positive control for 100% cytotoxicity. Each experiment was performed in triplicate.

### 2.7. Membrane Depolarization

The evaluation of membrane potential was conducted by employing the voltage-sensitive fluorescent dye diSC_3_-5, as described previously [40]. Gram-positive *S. aureus* cells were centrifuged after reaching the mid-log point at 37 °C, corresponding to an OD600 of 0.5. Subsequently, the bacteria were resuspended in a washing buffer at OD600 nm = 0.05, following two washes with the same buffer solution containing 20 mM glucose and 5 mM HEPES at pH 7.4. Upon adding 20 nM diSC_3_-5 to the cell suspension, the dye was allowed to fully absorb into the bacterial membrane until a stable fluorescence value was achieved. The intensity of fluorescence emitted from diSC_3_-5 (excitation = 622 nm, emission = 670 nm) increased following the addition of peptides, serving as an indicator of membrane depolarization. The addition of 0.1% Triton X-100 completely abolished the membrane potential.

### 2.8. SYTOX Green Uptake Assay

The SYTOX green experiment was utilized to assess the impact of peptides on bacterial membrane permeabilization [41]. *S. aureus* cells in the mid-log phase (OD600 = 0.5) were washed three times in a buffer containing 20 mM glucose and 5 mM HEPES (pH 7.4). Subsequently, the bacterial cell suspensions were diluted in a buffer consisting of 5 mM HEPES, 100 mM KCl, and 20 mM glucose, at pH 7.4, to a concentration of 1 × 10^6^ CFU/mL. After incubating for fifteen minutes in complete darkness, 0.5 μM SYTOX green was added to the bacterial suspensions. The Shimadzu fluorescence spectrophotometer RF-5300PC (Shimadzu Scientific Instruments, Kyoto, Japan) was employed to monitor SYTOX green fluorescence following the addition of peptides at a concentration of 2 × MIC. In the Shimadzu fluorescence spectrophotometer, the excitation and emission wavelengths were set at 485 nm and 520 nm, respectively.

### 2.9. Membrane Permeability Assay

The fluorescent probe NPN (1-N-phenylnaphthylamine) was utilized to measure the outer membrane permeability of Gram-negative *E. coli* [38]. Mid-log phase *E. coli* cells were diluted to an OD600 of 0.05 after being washed three times in a buffer containing 5 mM HEPES, 20 mM glucose, and 5 mM KCN at pH 7.4. A 1 mM stock solution of NPN was prepared by dissolving NPN in acetone. To achieve a final concentration of 10 µM, 30 μL of the stock solution was added to the bacterial suspension. The fluorescence was observed with an emission wavelength (λ) of 420 nm and an excitation wavelength (λ) of 350 nm until stable fluorescence was reached. Fluorescence was then measured over time after the peptides were added, with the concentration increasing until it reached a steady state. ONPG (o-nitrophenyl-β-galactosidase), a non-chromogenic substrate used for cytoplasmic membrane β-galactosidase enzyme, was employed to assess the release of β-galactosidase from *E. coli* ML-35, aiming to determine the inner membrane permeability of peptides [42]. A 1.5 mM solution of ONPG was added to a sample buffer consisting of 10 mM sodium phosphate and 100 mM NaCl at pH 7.4, which was used to suspend mid-log phase *E. coli* ML-35 bacteria to an OD600 of 0.5. Spectrophotometry at 405 nm was used to assess peptide-induced membrane permeabilization. The fluorescence of ONPG increases when it is hydrolyzed to form o-nitrophenol. The permeability of the inner membrane of *E. coli* ML-35 bacteria was assessed by the influx of ONPG, a substance that was then broken down into o-nitrophenol, forming a yellow product, by the presence of β-galactosidase in the cytoplasm.

### 2.10. Antibiofilm Activity

We assessed the antibiofilm activity of the peptides by determining their minimum biofilm eradication concentration (MBEC) and minimum biofilm inhibition concentration (MBIC) against drug-resistant bacteria, including MDRPA and MRSA, as previously described [43,44]. A 96-well plate containing a subculture of 1 × 10^6^ CFU/200 μL of bacteria was incubated overnight at 37 °C with or without peptides to test for biofilm inhibition and eradication. The untreated culture served as the negative control, and LL-37 was used as the control peptide.

### 2.11. Confocal Laser Scanning Microscopy

Drug-resistant bacteria, including MDRPA and MRSA, were cultured to a concentration of 1 × 10^6^ CFU/mL in 24-well plates, and discs were submerged in MHB supplemented with glucose for a full day to facilitate biofilm formation [37]. Subsequently, discs containing planktonic cells were transferred to new 24-well plates containing peptides, following three washes with PBS. The plates were then incubated for 6 h. Afterward, the discs were removed, washed twice with PBS, and stained simultaneously with 40 μM propidium iodide (PI) and 6.7 μM SYTO 9. The biofilm mass was visualized as a planar image using confocal laser scanning microscopy (ZEISS Microscopy LSM 710 Meta, Jena, Germany) and analyzed using the ZEN 2009 Light Edition software, version 4.2.0.121, following a 30 min dark incubation period at 37 °C.

### 2.12. FACScan Analysis

A flow cytometer was utilized to assess bacterial membrane integrity by quantifying the uptake of PI by the cells [37]. In brief, bacterial cells in the mid-log phase of MRSA and MDRPA were diluted to achieve an OD600 of 0.5. An equal volume of PBS and cell suspension was added to the mixture, which was then centrifuged at 8000 RPM for 5 min. The resulting cell pellets were resuspended in PBS. Subsequently, 10 µL of PI was added, and the mixture was incubated for 15 min. Peptides at a concentration of 2 × MIC were then added to the mixture, followed by an additional fifteen-minute incubation. A FACScan device (Agilent, Santa Clara, CA, USA) was used to quantify PI fluorescence.

## 3. Results and Discussion

### 3.1. Peptide Design and Characterization

Melittin contains four leucine residues at positions 6, 9, 13, and 16. In this study, we systematically substituted four leucine residues (L6, L9, L13, and L16) in melittin with 6-aminohexanoic acids, aiming to alter the leucine zipper motif and enhance the structural flexibility of melittin. Specifically, Mel-LX1, Mel-LX2, Mel-LX3, and Mel-LX4 represent individual substitutions of leucine residues at positions 6, 9, 13, and 16 with 6-aminohexanoic acid, respectively. Meanwhile, Mel-LX5 serves as an analog with simultaneous substitution of all four leucine residues.

The synthesized melittin and its analogs underwent mass spectroscopic examination, precisely determining the molecular weight of the peptides (Appendix A). Despite the identical molecular weights of the peptides, analysis by RP-HPLC revealed distinct changes in retention times (Appendix A). The results obtained from RP-HPLC and mass spectroscopy for melittin and its analogs indicated that the peptides were synthesized with adequate purity and precision. Table 1 provides the sequences, HPLC retention times, net charges, and mass analyses of melittin and its analogs.

### 3.2. CD Spectroscopy

CD spectroscopy was used to analyze the secondary structure of melittin and its analogs in an aqueous solution and in membrane-mimicking environments (30 mM SDS and 50% TFE), as shown in Figure 1. In an aqueous solution, melittin molecules are known to be monomeric and have a random coil structure at low concentrations, whereas at high concentrations, melittin folds into α-helical tetramers. The CD spectra of melittin at 100 μg/μL concentration in a 10 mM sodium phosphate buffer showed characteristic α-helical patterns, with molar lower mean residue ellipticity at 208 nm and 222 nm, whereas all its analogs (Mel-LX1, Mel-LX2, Mel-LX3, Mel-LX4, and Mel-LX5) exhibited a characteristic disordered structure, with the lowest point occurring at about 198 nm. In our conditions, melittin appears to be a tetramer in aqueous buffer, and the analogs are monomers. When exposed to a solution containing 30 mM SDS and 50% TFE, all peptides displayed two distinct negative bands at 208 and 222 nm, indicating that the peptides mostly adopted stable α-helical conformations in the membrane-mimicking environment. Compared to melittin, Mel-LX1, Mel-LX2, and Mel-LX4 showed relatively lower intensity, and Mel-LX5 had the lowest, suggesting that the incorporation of 6-aminohexanoic acid provides structural flexibility in the lipid bilayers. Mel-LX3 displayed comparable levels of α-helical structure to melittin, suggesting that both Mel-LX3 and melittin are likely to interact with membranes in a similar manner. It is noteworthy that the proline at position 14 of melittin introduces a kink of an α-helix. The substitution of leucine at position 13 for 6-aminohexanoic acid appears to have less effect on structural changes compared to other positions.

### 3.3. Antibacterial Activity of Melittin and Its Analogs

The antibacterial activity of melittin and its analogs was assessed against four Gram-positive bacteria, including MRSA, and four Gram-negative bacteria, including MDRPA. Melittin demonstrated antibacterial activity, with MIC values ranging from 4 to 16 µM against all tested strains. In comparison to melittin, Mel-LX1, Mel-LX2, and Mel-LX4 displayed 4- to 16-fold lower activity, while Mel-LX5 was almost inactive. However, Mel-LX3 was similarly active as melittin against both Gram-positive and Gram-negative bacteria. Importantly, Mel-LX3 effectively inhibited the growth of drug-resistant bacteria, including MRSA and MDRPA. The potent antibacterial activity of Mel-LX3 suggests that the leucine at position 13 contributes less significantly to its effectiveness against bacterial strains.

### 3.4. Hemolytic and Cytotoxic Activities of Melittin and Its Analogs

We then examined the hemolytic impact of the peptides on sRBCs, as depicted in Figure 2a. At a concentration of 2 μM, melittin induced nearly 100% hemolysis, whereas all melittin analogs exhibited minimal hemolytic activity of less than 5%. Even at a concentration of 64 µM, hemolysis by Mel-LX1, Mel-LX2, Mel-LX3, Mel-LX4, and Mel-LX5 was 11%, 6%, 27%, 3%, and 2%, respectively. We further assessed the cytotoxicity of the peptides against RAW 264.7 macrophage cells, as depicted in Figure 2b. Melittin resulted in less than 10% cell survival rate at 4 µM, whereas all analogs displayed over 80% cell survival at the same concentration. These findings suggest that regardless of the leucine position in melittin, substitution of the leucine residue with 6-aminohexanoic acid significantly diminishes its hemolytic and cytotoxic effects. These results align with prior research indicating the crucial role of leucine residues in the hemolytic activity and cytotoxicity of melittin [30,45]. The hemolytic activity of melittin is closely related to its hydrophobicity, and its oligomerization in aqueous buffer is likely driven by these hydrophobic interactions.

To evaluate cell selectivity towards bacterial and mammalian cells, the therapeutic index (TI) was calculated based on antibacterial and hemolytic activities (Table 2). The TI was determined as the ratio of the 10% hemolysis (HC_10_) to the geometric mean (GM) of the MIC. A higher TI indicates better cell selectivity. Among the melittin analogs, Mel-LX3 exhibited the highest TI value of 5.12, significantly higher than melittin’s TI of 0.19. Other analogs, such as Mel-LX1, Mel-LX2, Mel-LX4, and Mel-LX5, showed TI values of 0.65, 2.28, 2.13, and 0.66, respectively. Notably, Mel-LX3 demonstrated the most enhanced selectivity for bacteria over mammalian cells.

### 3.5. Membrane Permeabilization of Gram-Positive Bacteria

Many AMPs with an amphipathic α-helical structure permeabilize the cytoplasmic membrane of Gram-positive bacteria, causing a breakdown in transmembrane potential, resulting in cell death [46]. We assessed the effects of melittin and Mel-LX3 on the membrane depolarization of Gram-positive *S. aureus* by monitoring the fluorescence intensity of diSC_3_-5, a dye that is sensitive to potential (Figure 3a). Buforin-2, known to be an intracellular target without membrane permeabilization, was used as a negative control. Both melittin and Mel-LX3 at two times the MIC induced rapid depolarization of the membrane against *S. aureus*. Both peptides caused depolarization of the cytoplasmic membrane, which increased fluorescence within 2 min. We then evaluated the membrane permeability of *S. aureus* using SYTOX green, which is a cationic dye that can only pass through compromised membranes (Figure 3b). Buforin-2 as a negative control did not affect fluorescence intensity, but both melittin and Mel-LX3 at 2 × MIC resulted in a rapid increase in fluorescence, reaching the highest levels within 2 min. These results imply that the primary antimicrobial mechanism of Mel-LX3 is related to membrane permeabilization of Gram-positive *S. aureus*.

### 3.6. Membrane Permeabilization of Gram-Negative Bacteria

Given the presence of both outer and inner membranes in Gram-negative bacteria, we examined the permeability of these membranes in *E. coli* induced by melittin and Mel-LX3. The outer membrane permeability of *E. coli* was evaluated using N-phenyl-1-naphthylamine (NPN) fluorescence, which exhibits weak fluorescence in the intact outer membrane but becomes more pronounced when the outer membrane is compromised (Figure 4a). The outer membrane permeability of *E. coli* increased in a dose-dependent manner in response to melittin and Mel-LX3. Notably, Mel-LX3 at a concentration below the MIC (2 µM) significantly permeabilized the outer membranes of *E. coli*, similar to melittin. No appreciable outer membrane permeability was observed when buforin-2 was applied. To analyze the inner membrane permeability of Gram-negative bacteria induced by the peptides, we measured the hydrolysis of ortho-nitrophenyl-β-D-galactosidase (ONPG) due to the release of cytoplasmic β-galactosidase from *E. coli* ML-35 (Figure 4b). We observed that fluorescence intensity increased with the concentration of melittin and Mel-LX3, indicating that the peptides induce inner membrane permeability in a dose-dependent manner. These results indicate that Mel-LX3 is capable of permeabilizing both the outer and inner membranes of Gram-negative bacteria, similar to melittin.

### 3.7. Flow Cytometric Analysis

A flow cytometric analysis was conducted to evaluate the membrane integrity of drug-resistant bacteria (MRSA and MDRPA) following treatment with melittin and Mel-LX3 (Figure 5). Propidium iodide (PI), a DNA intercalating dye, was used as an indicator to assess membrane integrity and cell death via flow cytometry. In the absence of peptides, minimal PI staining was observed in the bacteria, indicating intact cell membranes. As expected, the negative control, buforin-2, resulted in less than 10% PI staining of the bacteria, suggesting that buforin-2 does not disrupt the bacterial membrane. In contrast, both melittin and Mel-LX3 led to significant PI staining, with melittin resulting in 98% and 97% PI staining of MRSA and MDRPA, respectively, and Mel-LX3 inducing 94% PI staining of both MRSA and MDRPA. These findings suggest that both peptides are capable of damaging the bacterial cell membrane. This analysis supports the hypothesis that Mel-LX3 damages bacterial cell membrane integrity, potentially contributing to its antimicrobial properties against these resistant bacteria.

### 3.8. Antibiofilm Activity

Biofilms represent a significant threat to environmental health as they enable bacteria to form resilient communities, leading to various diseases and fostering antibiotic resistance [47,48]. We investigated whether melittin and Mel-LX3 can inhibit the growth of biofilms of MRSA and MDRPA, as well as eradicate preformed biofilms. LL-37, known for its antibiofilm properties, was used as a control. Figure 6 illustrates the antibiofilm activity of melittin and Mel-LX3 against drug-resistant bacteria, including MRSA and MDRPA. When exposed to the peptides, LL-37 inhibited biofilm formation by approximately 50% at concentrations ranging from 8 to 16 µM. In contrast, melittin and Mel-LX3 effectively inhibited nearly 90% of biofilm formation at concentrations of 4 µM for MRSA and 8 µM for MDRPA (Figure 6a,b). Moreover, both melittin and Mel-LX3 demonstrated nearly 90% MBEC at 64 µM for MRSA and 8 µM for MDRPA (Figure 6c,d). Particularly noteworthy is their significantly stronger biofilm eradication activity against MDRPA compared to LL37. We then employed confocal laser scanning microscopy to visualize the impact on live/dead biofilm bacteria (Figure 6e,f). Live cells were stained with a green fluorescence dye (SYTO-9), while dead cells were stained with a red fluorescence dye (PI). The treatment of biofilm-formed MRSA and MDRPA with melittin or Mel-LX3 resulted in a notable decrease in the number of live cells and a significant increase in the number of dead cells. This suggests that both melittin and Mel-LX3 are effective in killing biofilm-formed MDRPA and MRSA.

## 4. Conclusions

This study synthesized five analogs of melittin by substituting leucine residues with the structural isomer 6-aminohexanoic acid with the aim of enhancing cell selectivity. Among the analogs, Mel-LX3 exhibited potent antibacterial activity against both Gram-positive and Gram-negative bacteria, comparable to melittin, while demonstrating significantly reduced hemolytic and cytotoxic effects. Mechanistic studies revealed that Mel-LX3 effectively permeabilized bacterial membranes, leading to its enhanced antibacterial efficacy. Particularly noteworthy is the ability of Mel-LX3 to combat MRSA and MDRPA, as well as its effectiveness in inhibiting and eradicating biofilms, especially in MDRPA strains. With its improved bacterial selectivity and anti-biofilm activity, Mel-LX3 emerges as a promising candidate for the development of new antimicrobial agents. The substitution of leucine with 6-aminohexanoic acid in AMPs represents a promising strategy for addressing resistant bacteria by reducing toxicity and maintaining antibacterial efficacy, offering potential solutions to combat infectious diseases in the future.

## Figures and Tables

**Figure 1 biomolecules-14-00699-f001:**
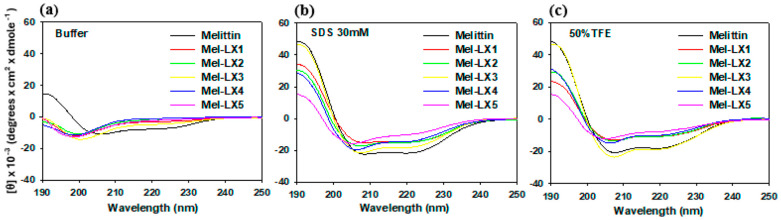
CD spectra of melittin and its analogs in (**a**) 10 mM sodium phosphate buffer, (**b**) 30 mM SDS, and (**c**) 50% TFE.

**Figure 2 biomolecules-14-00699-f002:**
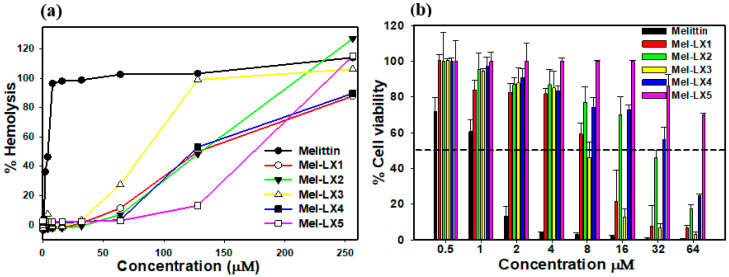
Hemolytic and cytotoxic activities of melittin and Mel-LX3. (**a**) Percentage of hemolysis of sRBCs induced by the peptides. (**b**) Percentage of cell viability in RAW264.7 mouse macrophages. Cell viability was assessed using an MTT dye reduction assay.

**Figure 3 biomolecules-14-00699-f003:**
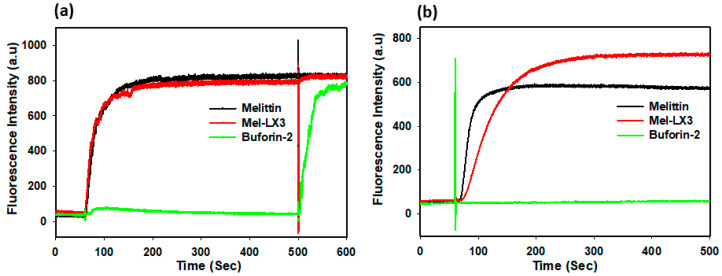
Membrane permeabilization of Gram-positive bacteria induced by melittin and Mel-LX3. (**a**) Membrane depolarization of *S. aureus*. Cytoplasmic membrane depolarization was measured using the fluorescent intensity of the membrane potential-sensitive dye DiSC_3_-5 when treated with melittin and Mel-LX3 at 2 × MIC. (**b**) SYTOX green entry due to membrane alterations. Increased fluorescence when treading the peptides at 2 × MIC indicates the entry of the SYTOX green probe into *S. aureus* cells.

**Figure 4 biomolecules-14-00699-f004:**
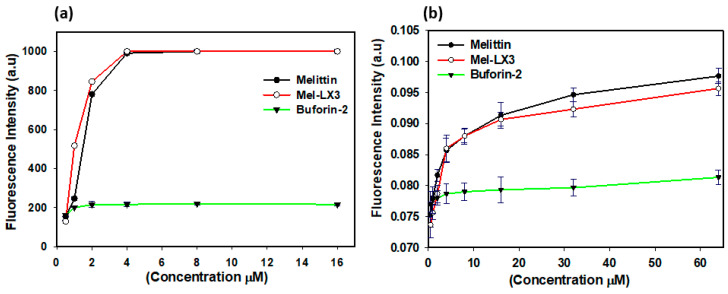
Membrane permeabilization of Gram-negative bacteria. (**a**) Permeabilization of outer membranes. NPN uptake in *E. coli* (KCTC 1682) was monitored in the presence of varying peptide concentrations. (**b**) Permeabilization of inner membranes. Hydrolysis of ONPG occurs due to the release of cytoplasmic β-galactosidase from *E. coli* ML-35 bacterial cells treated with peptides at different concentrations.

**Figure 5 biomolecules-14-00699-f005:**
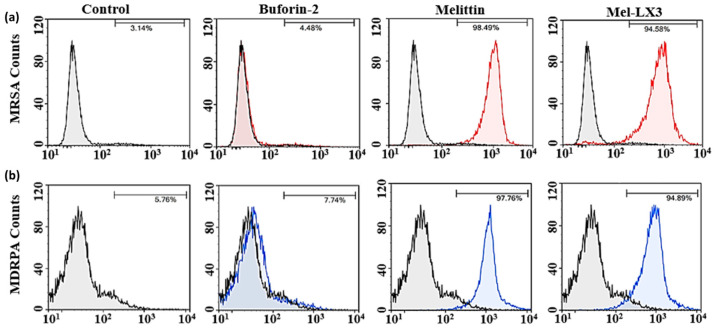
Membrane integrity of drug-resistant bacteria. Flow cytometry was used to monitor membrane integrity of (**a**) MRSA and (**b**) MDRPA in the absence and presence of peptides.

**Figure 6 biomolecules-14-00699-f006:**
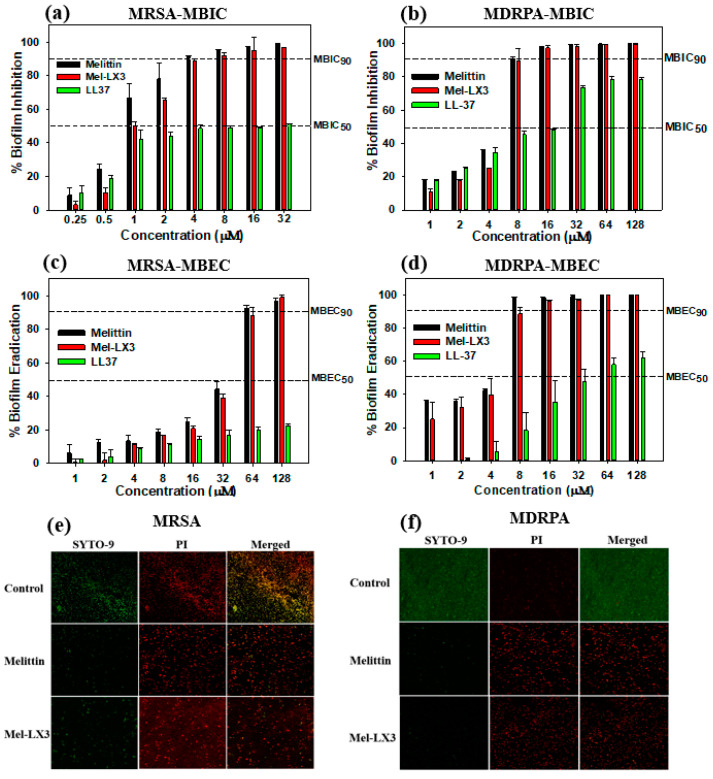
Antibiofilm activity of melittin and Mel-LX3 against drug-resistant bacteria. (**a**) Biofilm inhibition by peptides against MRSA. (**b**) Biofilm inhibition by peptides against MDRPA. (**c**) Biofilm eradication by peptides against MRSA. (**d**) Biofilm eradication by peptides against MDRPA. The dashed lines represent the 50% and 90% inhibition and eradication concentrations. Visualization of antibiofilm properties of the peptides against (**e**) MRSA and (**f**) MDRPA. Live and dead cells are illustrated in green (SYTO 9) and red (PI) fluorescence, respectively.

**Table 1 biomolecules-14-00699-t001:** Amino acid sequences and physicochemical properties of melittin and its analogs.

Peptides	Sequences	Rt ^a^ (min)	Net Charge	Mass Analysis ^b^		
				Mass (g/mol)	m/z Calculated	m/z Observed
Melittin	GIGAVLKVLTTGLPALISWIKRKRQQ	30.64	6	2847.49	949.83	949.8
Mel-LX1	GIGAV**X**KVLTTGLPALISWIKRKRQQ	24.93	6	2847.49	949.83	949.8
Mel-LX2	GIGAVLKV**X**TTGLPALISWIKRKRQQ	24.76	6	2847.49	949.83	949.7
Mel-LX3	GIGAVLKVLTTG**X**PALISWIKRKRQQ	26.80	6	2847.49	949.83	949.7
Mel-LX4	GIGAVLKVLTTGLPA**X**ISWIKRKRQQ	24.75	6	2847.49	949.83	949.7
Mel-LX5	GIGAV**X**KV**X**TTG**X**PA**X**ISWIKRKRQQ	18.76	6	2847.49	949.83	949.7

^a^ Retention time (Rt) was evaluated by RP-HPLC with C_18_ reversed-phase column. ^b^ Molecular masses were analyzed by ESI-MS. m/z: mass-to-charge ratio of [M+3H]^3+^. The X shown in bold indicates 6-aminohexanoic acid.

**Table 2 biomolecules-14-00699-t002:** Antibacterial activities of melittin and its analogs against Gram-positive and Gram-negative bacteria.

Microorganism	MIC (μM)					
	Melittin	Mel-LX1	Mel-LX2	Mel-LX3	Mel-LX4	Mel-LX5
Gram-positive organisms						
*S. aureus* (KCTC 1621)	8	64	32	8	32	>128
*S. epidermidis* (KCTC 1917)	8	32	32	16	32	>64
*B. subtilis* (KCTC 3068)	16	16	16	16	32	>64
*MRSA* (CCARM 3090)	4	>128	128	4	128	>128
Gram-negative organisms						
*E. coli* (KCTC 1682)	8	64	32	16	32	>128
*P. aeruginosa* (KCTC 1637)	16	64	64	16	64	>64
*S. typhimurium* (KCTC 1926)	16	32	16	16	32	>64
*MDRPA* (CCARM 2095)	8	>128	128	8	128	>128
GM (μM) ^a^	10.5	98	56	12.5	60	192
HC_10_ (μM) ^b^	2	64	128	64	128	128
TI (HC_10_/GM) ^c^	0.19	0.65	2.28	5.12	2.13	0.66

^a^ Geometric mean (GM) of the minimum inhibitory concentration (MIC). ^b^ HC_10_ is the minimum inhibitory concentration that causes 10% hemolysis of sRBC. ^c^ Therapeutic index (TI) is the ratio of HC_10_/GM.

## Data Availability

Data are contained within this article and the Appendix A.

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
