# Peer review of "Enhancing Selective Antimicrobial and Antibiofilm Activities of Melittin through 6-Aminohexanoic Acid Substitution"

_biomolecules, 2024, doi:10.3390/biom14060699_

Round 1

Reviewer 1 Report

Comments and Suggestions for Authors

See attached file.

Reviewer 2 Report

Comments and Suggestions for Authors

Authors presented studies on the chemical modification of Melittin, the main components of the bee venom, but also possessing substantial antibacterial activity. The main objective of the work was to improve antibacterial activity by substitution of leucine with 6-amino hexanoic acid. In general hypothesis was negatively verified, since none of new derivatives were more active then melittin. However one derivative display similar activity but is less toxic. Work is intersting but requires improvements before publication.

- The change in the peptide structure is substantial compare to melitilne, since alfa amino acid is replaced with epsilon-amino acid. What was motivation for such a change?

- Material synthesis  and products were not sufficiently described. Please provide evidence that you are dealing with molecules you intend to have. Showing identical molecular ion for all derivative is not a proof (table  1). Fragmentation of molecular ion and other ions should be presented and discussed to show differences between investigated molecules. Ideally NMR spectroscopy data should be supplemented.

- Some conclusion should be rephrased since all derivatives are not more active than melittin. E.g. in the abstract authors write: “We propose that the substitution of leucine with 6-aminohexanoic acid in AMPs represents a significant strategy in combating resistant bacteria.” Later in conclusions:” The substitution of leucine with 6-aminohexanoic acid in AMPs 441 represents a significant strategy in addressing resistant bacteria, offering potential 442 solutions to combat infectious diseases in the future.”

- Additionally what is the meaning of the phrase :”significant strategy“ ?

- At page 6 authors write: “ Despite the similar molecular weights of the peptides, …” mass of modifications are identical as are molecular ions (table 1), not similar, fragmentation ions should be different.

Comments on the Quality of English Language

Fig 5 caption reads: “obsence” should read “absence”

Round 2

Reviewer 2 Report

Comments and Suggestions for Authors

Authors response clarified my concerns